# Dissecting the precise role of H3K9 methylation in crosstalk with DNA maintenance methylation in mammals

Qian Zhao[1],[*], Jiqin Zhang[1],[*], Ruoyu Chen[1],[*], Lina Wang[1],[*], Bo Li[1], Hao Cheng[2], Xiaoya Duan[1], Haijun Zhu[1], Wei Wei[1], Jiwen Li[1], Qihan Wu[1], Jing-Dong J. Han[2], Wenqiang Yu[3], Shaorong Gao[4], Guohong Li[5] & Jiemin Wong[1],[6]

In mammals it is unclear if UHRF1-mediated DNA maintenance methylation by DNMT1 is strictly dependent on histone H3K9 methylation. Here we have generated an *Uhrf1* knockin (KI) mouse model that specifically abolishes the H3K9me2/3-binding activity of Uhrf1. The homozygous *Uhrf1* KI mice are viable and fertile, and exhibit ∼10% reduction of DNA methylation in various tissues. The reduced DNA methylation occurs globally in the genome and does not restrict only to the H3K9me2/3 enriched repetitive sequences. *In vitro* UHRF1 binds with higher affinity to reconstituted nucleosome with hemi-methylated CpGs than that with H3K9me2/3, although it binds cooperatively to nucleosome with both modifications. We also show that the nucleosome positioning affects the binding of methylated DNA by UHRF1. Thus, while our study supports a role for H3K9 methylation in promoting DNA methylation, it demonstrates for the first time that DNA maintenance methylation in mammals is largely independent of H3K9 methylation.

[1] Shanghai Key Laboratory of Regulatory Biology, The Institute of Biomedical Sciences and School of Life Sciences, East China Normal University, Shanghai 200241, China. [2] Key Laboratory of Computational Biology, CAS Center for Excellence in Molecular Cell Science, Collaborative Innovation Center for Genetics and Developmental Biology, Chinese Academy of Sciences-Max Planck Partner Institute for Computational Biology, Shanghai Institutes for Biological Sciences, Chinese Academy of Sciences, 320 Yue Yang Road, Shanghai 200031, China. [3] Department of Biochemistry and Molecular Biology, Laboratory of RNA Epigenetics, Institutes of Biomedical Sciences, Shanghai Medical College, Fudan University, 130 Dong-An Road, Shanghai 200032, China. [4] Clinical and Translational Research Center of Shanghai First Maternity, Infant Hospital, School of Life Sciences and Technology, Tongji University, Shanghai 200092, China. [5] National Laboratory of Biomacromolecules, Institute of Biophysics, Chinese Academy of Sciences, Beijing 100101, China. [6] Collaborative Innovation Center for Cancer Medicine, Sun Yat-Sen University Cancer Center, Guangzhou 510060, China. * These authors contributed equally to this work. Correspondence and requests for materials should be addressed to G.L. (email: liguohong@sun5.ibp.ac.cn) or to J.W. (email: jmweng@bio.ecnu.edu.cn).

DNA methylation in cytosine is a conserved epigenetic modification that plays important roles in transcriptional regulation and genome stability[1–3]. In mammals, DNA methylation occurs predominantly in CG dinucleotides and is required for embryonic development, transcriptional regulation, heterochromatin formation, X-inactivation, imprinting and genome stability[3–5]. How the patterns of DNA methylation are established and maintained has been the central question for epigenetic study.

Histone modifications have critical roles in regulation of chromatin structure and function[6,7]. As two major mechanisms for epigenetic regulation, DNA methylation and histone modifications must act coordinately and growing evidence supports crosstalk between these two types of modification[1,6,8,9]. For examples, studies from *Neurospora* have demonstrated that in this model organism the DNA methylation catalysed by DIM-2 is strictly dependent on a histone H3K9 methyltransferase DIM-5 (refs 10,11). Mechanistically, DIM-5 catalysed H3K9 methylation is recognized by a mammalian HP1 homologue protein, which interacts directly with DIM-2 and is required for recruiting DIM-2 to methylate DNA[12]. Although DNA methylation in plant *Arabidopsis* is more complicated and involves more DNA methyltransferases, accumulative evidence indicates that the CMT2–CMT3 DNA methylation pathway and RNA-dependent DNA methylation pathway are directly or indirectly dependent on H3K9 methylation, respectively[1,13–16]. These studies from model organisms thus raise an intriguing question whether a H3K9 methylation-dependent DNA methylation pathway also exists in mammals.

In mammals the crosstalk between DNA methylation and histone methylation appears to be more complex. Three active DNA methyltransferases DNMT3A, DNMT3B and DNMT1 are responsible for all known CG and non-CG methylation in mammals[3,17]. DNMT3A and DNMT3B form complexes with enzymatically inactive DNMT3L and play critical roles in establishment of patterns of DNA methylation during gametogenesis and early embryonic development. DNMT3L and DNMT3A have been shown to connect unmethylated H3K4 with *de novo* DNA methylation[18,19]. Suv39h-mediated H3K9 methylation has been reported to direct Dnmt3b through a HP1α-Dnmt3b interaction to methylate major satellite repeats at pericentric heterochromatin[20]. In addition, a recent study has revealed a role for H3K36 methylation in targeting DNMT3B for genic DNA methylation[21]. In mammals the patterns of DNA methylation in somatic cells are to a large extent maintained by the activity of DNMT1. The direct interaction between DNMT1 and G9a has been proposed to coordinate H3K9 methylation and DNA methylation during DNA replication[22], although it is not clear to what extent this interaction mediates the crosstalk of H3K9 and DNA methylation. Despite of aforementioned crosstalk between DNMT1 and Dnmt3b with H3K9 methylation, UHRF1 has emerged from recent studies as an accessory protein essential for DNA maintenance methylation by DNMT1 (refs 23,24) and the key player that mediates the crosstalk between H3K9 and DNA methylation in mammalian cells[25].

UHRF1, also known as ICBP90 in human and NP95 in mouse, is a multi-structural domain and functional protein. UHRF1 contains a SET and RING-associated (SRA) domain that specifically recognize hemi-methylated CpG and a Tandem Tudor domain (TTD) and a plant homeodomain (PHD) domain that bind cooperatively the H3 tails with H3K9me2/3 (refs 26–32). On the basis of the unique property of SRA, UHRF1 was initially proposed to recruit DNMT1 to replication forks by binding hemi-methylated CpGs generated during DNA replication[23,24]. The finding that UHRF1 also contains a specific TTD domain for binding H3K9me2/3 raises the question whether UHRF1 also mediates a H3K9 methylation-dependent DNA methylation[25,29]. On the basis of knockdown of endogenous UHRF1 and subsequent rescue with mutant UHRF1, two studies concluded that the H3K9me2/3-binding activity is essential for UHRF1 to mediate DNA methylation by DNMT1[33,34]. However, based on rescue experiment in *Uhrf1*$^{-/-}$ mouse embryonic stem (ES) cells, we reported previously that the H3K9me2/3-binding activity of UHRF1 promotes but is not absolutely essential for its ability to mediate DNA maintenance methylation[35]. Thus, it is currently unclear whether DNA maintenance methylation mediated by UHRF1 in mammalian cells is strictly dependent on H3K9 methylation, as the case of *Neurospora*.

In this study we have generated an *Uhrf1* knockin (KI) mouse model with specific impairment of Uhrf1's H3K9me2/3 binding activity. The homozygous *Uhrf1* KI mice are viable and fertile and exhibit ~10% reduction of DNA methylation in various tissues tested. Mechanistic study with *in vitro* reconstituted nucleosomes demonstrates that UHRF1 binds nucleosome with hemimethylated DNA better than that with H3K9me2/3, although it binds cooperatively to nucleosome with both H3K9me2/3 and hemimethylated DNA. Altogether our study provides compelling evidence that H3K9 methylation promotes but is not required for bulky DNA maintenance methylation in mammals. Thus, the mechanisms underlying the crosstalk between histone modifications and DNA methylation in mammalian cells are distinct from that in *Neurospora* and plants.

## Results

**Generation of *Uhrf1* YP187/188AA KI mice**. UHRF1, also known as ICBP90 and NP95, was previously identified as an H3K9me2/3-specific binding protein in our laboratory through a unbiased biochemical affinity-purification of HeLa nuclear proteins using immobilized H3K9me2 peptides[29]. Subsequent biochemical and structural studies collectively demonstrated that while the TTD and PHD domain in isolation is capable of binding H3K9me2/3 and unmodified H3R2 (refs 30,35–37), respectively, they act cooperatively to bind the histone H3 tail with H3K9me2/3 with high affinity[31,32]. A double amino acid mutation, namely tyrosine 191 and proline 192 in the TTD domain of human UHRF1 has been shown to completely impair the binding of H3K9me2/3 by UHRF1 both *in vitro* and in cells without affecting its hemi-methylated DNA binding activity[35]. To define the physiological function of H3K9me2/3-binding activity of UHRF1 in development and DNA methylation, we decided to introduce KI mutations into the mouse *Uhrf1* gene by converting Uhrf1 tyrosine 187 and proline 188, which are equivalent to human UHRF1 Y191 and P192, to alanines. We generated mouse ES cell line with YP187/188AA KI mutation through homologous recombination (Fig. 1a). We then derived chimeric mice from the KI ES cells, which transmitted successfully the mutant allele. The wild-type (WT), heterozygous and homozygous KI mice were genotyped by polymerase chain reaction (PCR) (Supplementary Fig. S1) and the homozygous mice were also verified by DNA sequencing as shown in Fig. 1b.

**The Uhrf1 YP187/188AA mutant mice are viable and fertile**. Genotyping of 360 newborn mice from independent heterozygous crosses indicated that 77 mice (21.4%) were homozygous to the mutation, close to the expected ratio of Mendelian inheritance (Fig. 1c). Both litter sizes and gender distributions were also close to the expected ratio (Fig. 1c). In addition, no gross differences in growth, body weight and life span were observed among the WT, heterozygous and homozygous KI mice (data not shown). Thus, in contrast to early embryonic lethality

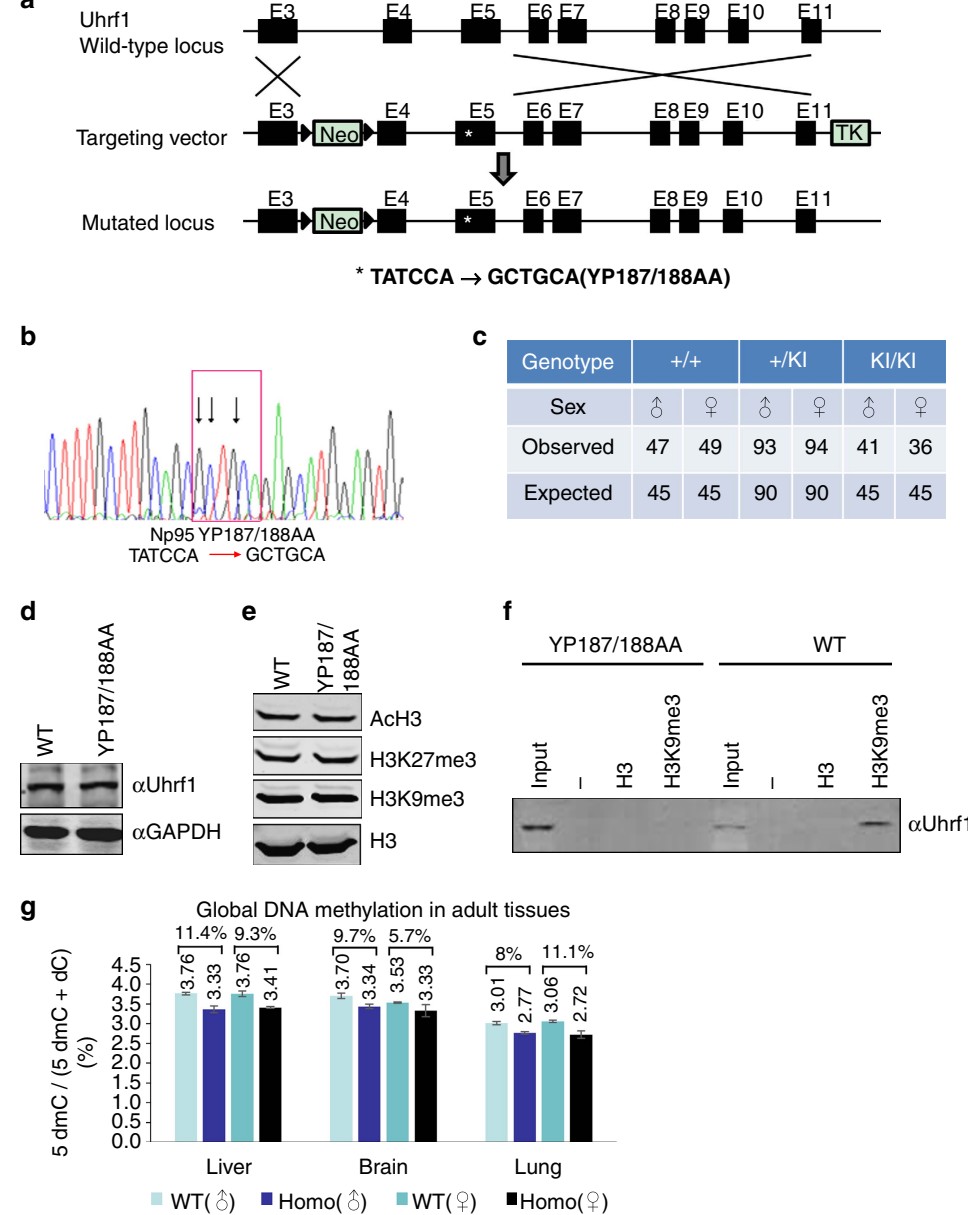

**Figure 1 | Generation and characterization of *Uhrf1* YP187/188AA KI mice.** (**a**) Diagram illustrating the scheme for generation of the *Uhrf1* YP187/188AA KI mice. The targeting vector contains Neo gene flanked by two FRT sites (triangles) and thymidine kinase (TK) gene for double selection. The asterisk represents the mutations of YP187/188AA (TATCCA→GCTGCA). (**b**) A representative sequencing data for homozygous YP187/188AA KI mice. (**c**) Summary of the genotyping results from the breeding of *Uhrf1* KI heterozygous mice. (**d**) Western blot result showing the levels of Uhrf1 proteins in liver tissues from the WT and KI mice. (**e**) Western blot analysis of core histones prepared from liver tissues of the WT and KI mice using antibodies as indicated. (**f**) *In vitro* pulldown assay confirmed that the Uhrf1 proteins from the liver tissues of KI mice were impaired in binding the histone N-terminal tail peptide with H3K9me3. (**g**) The global levels of DNA methylation in various mouse tissues derived from the WT and homozygous KI mice determined by HPLC analysis. Tissues were derived from three pairs of WT and KI littermates. The resulting genomic DNAs were pooled together for HPLC analysis. The level of methylation was shown as the percentage of 5dmC to 5dmC + 5dC.

phenotype of the *Uhrf1* straight knockout mice[24], the *Uhrf1* YP187/188AA KI mice are viable and fertile and display no obvious phenotype.

As expected from the lack of gross phenotypes for homozygous KI mice, we confirmed by western blot analysis of protein extracts derived from liver tissues of the littermates of WT and homozygous KI mice that the expression of Uhrf1 was unaltered in the KI mice (Fig. 1d). Similarly, we analysed core histones prepared from liver tissues of littermate WT and KI mice by western blot and observed no significant difference in the levels of

histone acetylation, H3K9me3 and H3K27me3 between WT and KI mice (Fig. 1e).

To test if the mutant Uhrf1 proteins in homozygous KI mice were indeed impaired in H3K9me2/3-binding activity, we prepared protein extracts from liver tissues of WT and homozygous KI mice. The subsequent peptide pulldown assay demonstrated that, while the Uhrf1 from the WT mice bound preferentially to the immobilized H3K9me3 peptide, the Uhrf1 proteins from the homozygous KI mice failed to do so (Fig. 1f). The *in vitro* pulldown assay thus demonstrated that, as expected,

the mutant Uhrf1 proteins were indeed defective in binding H3K9me2/3.

**Reduced global DNA methylation in YP187/188AA mice.** We next determined if the YP187/188AA mutation has any effect on global DNA methylation in KI mice. We prepared genomic DNA from liver, brain and lung tissues of the littermate WT and homozygous KI mice. To minimize the potential variation of individual mouse and gender difference, genomic DNAs were prepared from three different pairs of male and female littermate mice and pooled together according to gender. The levels of global DNA methylation were then measured by quantitative high-performance liquid chromatography (HPLC) assay. As shown in Fig. 1g, the results from three independent experiments showed that the levels of global DNA methylation presented as percentage of 5dmC versus 5dmC + dC were all lower in tissues from KI mice than the WT mice. The reduction of DNA methylation ranged from 5.7% in female brains to 11.4% in male livers. Thus, the *Uhrf1* YP187/188AA KI mice exhibit a widespread but moderate reduction of the global DNA methylation.

**A supportive role for H3K9 methylation in DNA methylation.** Having established that the loss of Uhrf1's H3K9me2/3-binding activity resulted in a moderate reduction of global DNA methylation, we wished to investigate precisely its effect on DNA methylation. In principle, the moderate reduction of global DNA methylation could be due to loss of DNA methylation in specific regions where DNA methylation is dependent on H3K9 methylation. Alternatively, it could be due to moderate reduction of DNA methylation in a large portion of the genome, reflecting a dominant role for hemi-methylated DNA in targeting Uhrf1 to replication forks via the function of its SRA domain. Thus, it is crucial to determine in a genome scale the DNA methylation patterns in the WT and KI mice. To this end, we carried out genome-wide RRBS analysis for pooled DNA from the liver tissues of three pairs of WT and KI littermate mice. RRBS reads were mapped to the mouse genome mm9 by bismark v0.12.2. As summarized in Fig. 2a, ∼60 million reads were obtained, which covered ∼7.2 million and ∼6.5 million CpG sites from the WT and KI mutant, respectively. Among them, 2.65 million CpGs were covered with at least 5 reads for both WT and KI mutant and were used for subsequent analysis. We found the mean level of CpG methylation in WT and KI mutant was 45.6% and 42.6%, respectively (Fig. 2b). This reflects an 8.7% reduction ((45.6–42.6%)/45.6%) of CpG methylation in KI mutant in comparison with the WT, which is close to ∼10% reduction of DNA methylation determined by HPLC analysis (Fig. 1g). Also, as shown in Fig. 2b, the patterns of DNA methylation in both WT and KI mutant show a typical biphasic distribution, with a shift to reduced DNA methylation in KI mutant. By defining differentially methylated CpGs as false discovery rate (FDR) < 0.05 and the absolute methylation difference > 25%, we identified in the KI mutant 20,308 hypomethylated CpGs and only 671 hypermethylated CpGs (Fig. 2a), which is consistent with a global reduction of DNA methylation in the KI mutant. Importantly, we found that the reduction of DNA methylation occurs broadly in the genome. If we simply separated the above CpG sites in WT and KI mutants into ten quantiles according to the levels of DNA methylation, it appears that reduction of DNA methylation was observed only in the category of the highest methylated CpGs (90–100% methylation) (Fig. 2c). However, if we observed the CpG sites in each of the ten quantiles and followed their changes of DNA methylation in the KI mutant, we found the levels of DNA methylation reduced in each quantile (Fig. 2d), with absolute methylation differences ranging from < 3% in the low

methylated sequences to ∼12% in the highest methylated regions. Furthermore, among the 2.65 millions of CpGs, only 2,512 CpGs (< 0.1%) were found to be highly methylated (> 70%) in WT and became lowly methylated (< 20%) in KI. Thus, the reduction of DNA methylation in KI mutants occurs broadly and does not appear to restrict only to the highly methylated CpGs. In support of this view, we found there is a 4–5% reduction of DNA methylation across the entire region in the mutant if the CpG sites were aligned according to their relative positions to the transcriptional start sites (Fig. 2e).

As H3K9me2/3 is known to be enriched in pericentromeric regions comprised of the major and minor satellite repetitive sequences and in repetitive sequences such as LINE[38–40], we looked specifically for the changes of DNA methylation in these and other repetitive sequences (Fig. 2f). We found even in these highly methylated repetitive sequences, the average relative reduction of DNA methylation was moderate and within the range of 8% (Fig. 2f), which is very close to the global level of reduction of DNA methylation (Fig. 2b). In fact, among in total 304,709 single repeat elements identified to have at least one CpG in our study, only 137 single repeat elements (< 0.04%) were highly methylated in WT (above 70%) and became poorly methylated in KI (below 20%). Thus, the reduction of CpG methylation in repetitive DNA sequences is not much more than CpG sites in the entire genome (Fig. 2b), indicating that DNA methylation in these H3K9me2/3-enriched regions is also not strictly dependent on the H3K9me2/3-binding activity of Uhrf1.

**H3K9 methylation enhances UHRF1 binding to nucleosome.** Previous studies have demonstrated that UHRF1 binds preferentially histone H3 tail containing H3K9me2/3 (refs 29–32). However, it has not been tested if UHRF1 also preferentially binds nucleosome with H3K9me2/3. In our previous study we found removal of the N-terminal UBL (ubiquitin-like domain) domain neither had effect on binding of H3K9me2/3 peptide nor heterochromatin localization in cells[35]. Furthermore, a recent study demonstrated that UHRF1 in HeLa nuclear extracts existed in an open conformation due to binding of PIP5 to a polybasic liner region between SRA and RING domain[41]. However, the full-length recombinant UHRF1 prepared from bacteria existed in a close conformation lacking the H3K9me2/3-binding activity due to an intramolecular interaction between the polybasic linker and TTD in the absence of PIP5 (ref. 41). To eliminate this intramolecular inhibition on UHRF1's H3K9me2/3-binding activity, we prepared glutathione *S*-transferase (GST)-fusion of recombinant UHRF1 with deletion of C-terminal polybasic linker plus the RING domain (aa 95–610) (Fig. 3a). We also prepared a mutant form of GST-UHRF1 with residues Y191 and P192 converted to A191 and A192 (YP191/192AA), which is equivalent to the YP187/188AA KI mutant in mouse Uhrf1 (Fig. 3a).

To analyse the binding of UHRF1 to nucleosomes *in vitro*, we reconstituted mononucleosomes with a 200 bp DNA fragment containing in the middle a 146 bp-strong 601 nucleosome positioning sequence[42] and an end-labelled biotin (Fig. 3b). Recombinant core histones were expressed and purified from bacteria and used for *in vitro* nucleosome reconstitution via salt dialysis[43]. We introduced H3K9me2 or H3K9me3 to H3 by chemical mimic approach[44]. A 6xHis tag was added to the N-terminus of H3 for the purpose of purification of chemically methylated H3. On nucleosome reconstitution with H3K9 unmethylated (H3K9me0), dimethylated (H3K9me2) or trimethylated (H3K9me3), the nucleosomes were purified by binding to streptavidin agarose beads and checked for histone components by SDS-PAGE followed by Commassie blue staining.

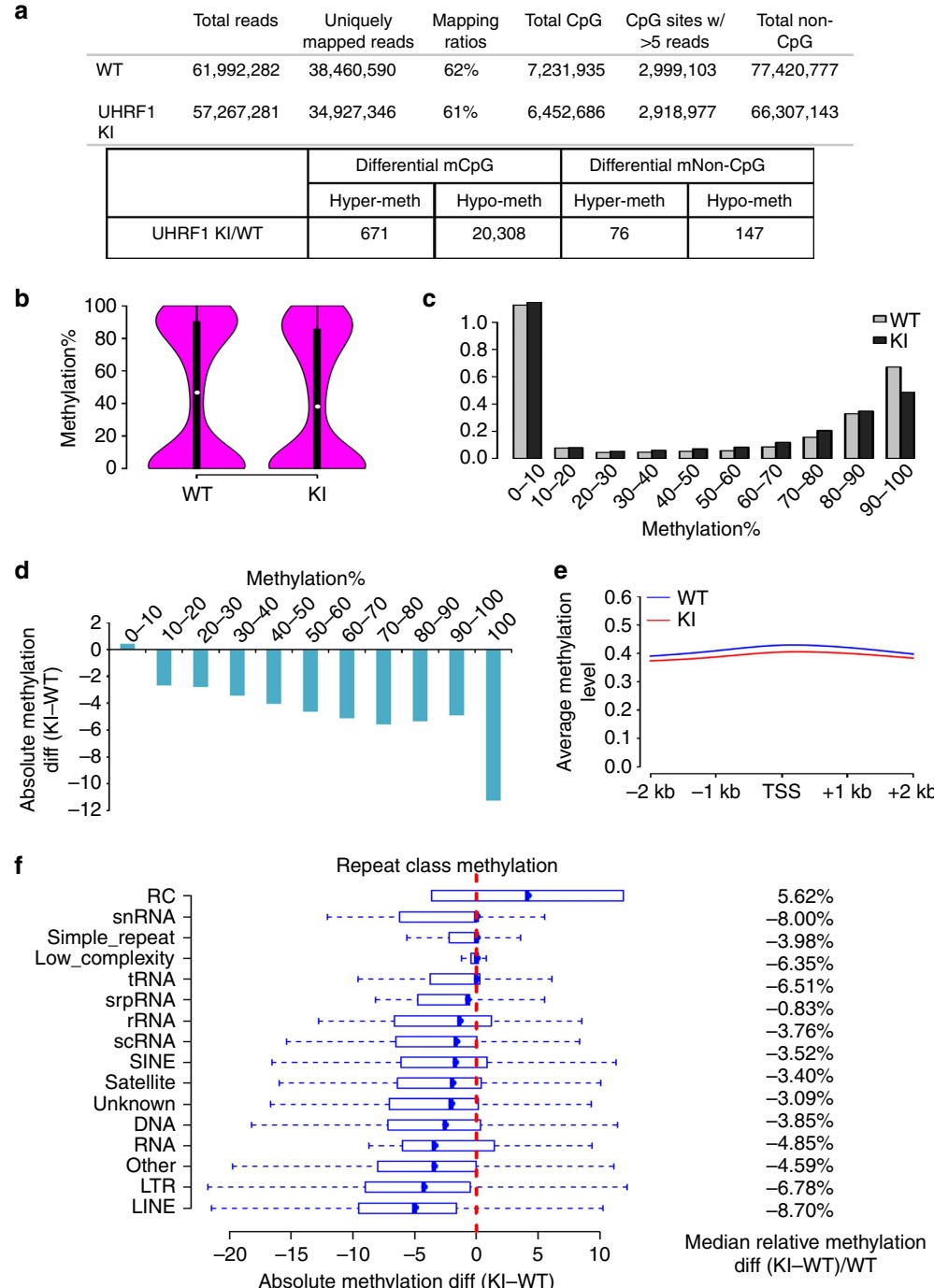

**Figure 2 | Genome-wide analysis of DNA methylation by RRBS demonstrates that the YP187/188AA mutation results in a global reduction of DNA methylation.** (**a**) RRBS reads were mapped to mouse genome mm9 by bismark v0.12.2. Only unique mapped reads were kept for further analysis. CpGs and non CpGs in both strands were reported. Differentially methylated CpGs were obtained by using R package methylkit and only the CpGs with at least 5 reads coverage were used for further analysis. Differentially methylated CpGs and non CpGs were defined by FDR < 0.05 and the absolute methylation difference above 25%. (**b**) 2,654,195 CpGs with at least 5 reads coverage in both WT and KI samples were plotted based on the percentages of DNA methylation. Mean CpG methylation levels in WT and KI are 45.6 and 42.6, respectively. Median CpG methylation levels in WT and KI are 46.7 and 38, respectively. Pair-wise t-test showed that the WT CpG methylation level was significantly higher than KI (P value < 0.001). (**c**) The distribution of CpGs with different percentage of methylation levels in WT and Uhrf1 KI. 2,654,195 CpGs covered by both samples were analysed here. X axis is the percentage of DNA methylation, Y axis is CpG counts in million. (**d**) The methylation difference between KI and WT for CpGs in each quantile was calculated and plotted in boxplot. Highly methylated CpG sites (above 70%) decreased more, but decrease of methylation was seen for each quantile of CpG methylation. But only 2,512 CpGs were found with high methylation levels in WT (above 70%) and low methylation levels in KI (below 20%). (**e**) Average CpG methylation levels in WT and KI from 2 kb upstream and 2 kb downstream of gene transcriptional start sites. (**f**) Methylation differences in various repeat sequences were shown in boxplots. Each box represents one repeat class methylation difference. 304,709 single repeat elements were used with at least one CpG covered. Only 137 single repeat elements were found with high methylation levels in WT (above 70%) and low methylation levels in KI (below 20%).

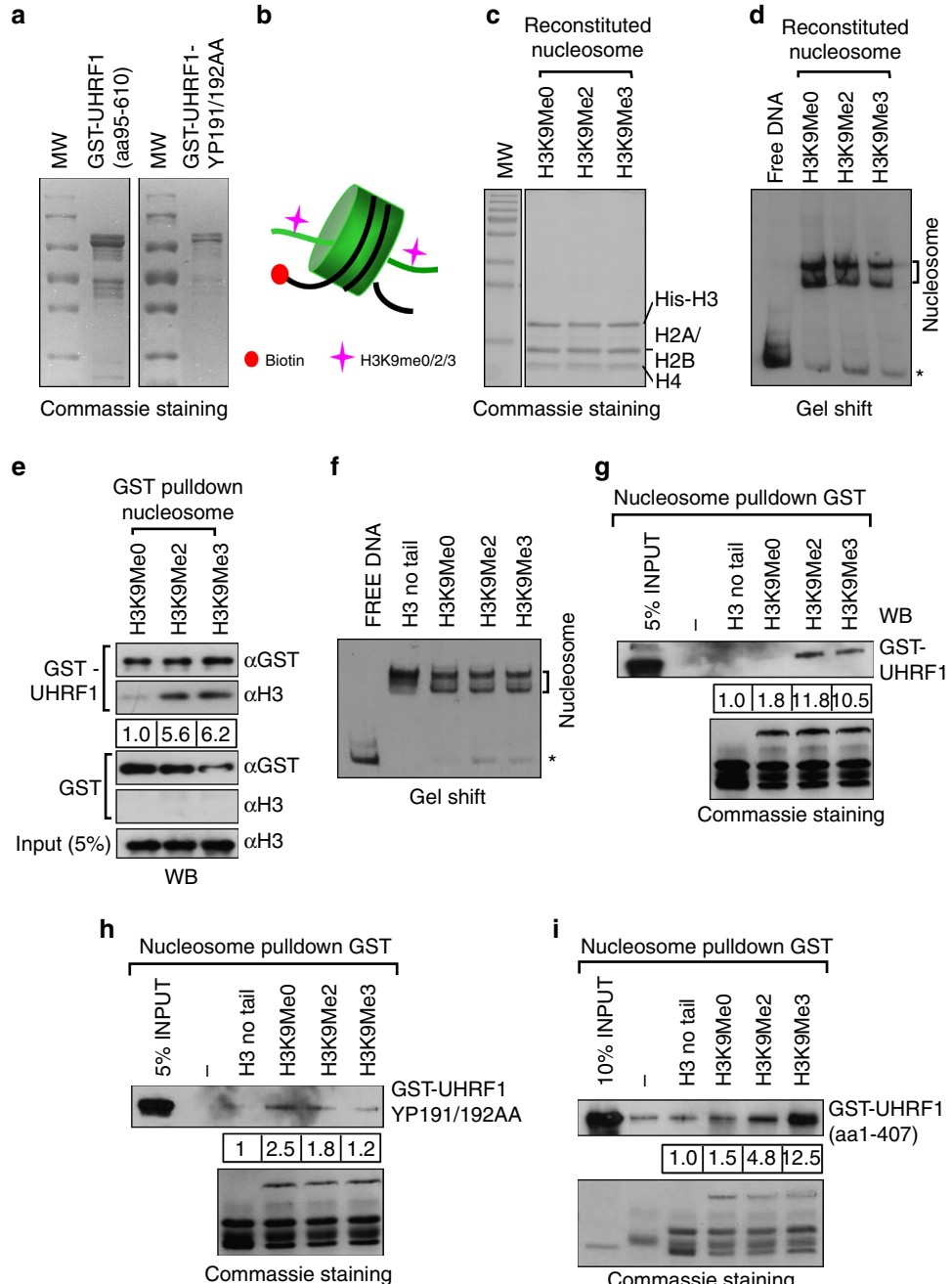

**Figure 3 | UHRF1 preferentially binds *in vitro* reconstituted nucleosome containing H3K9me3.** (**a**) Commassie blue staining gel showing recombinant UHRF1 fusion proteins. The recombinant GST-UHRF1 (aa 95–610) and its YP191/192AA mutant were expressed and purified from *E.coli*. The YP191/192AA mutation in human UHRF1 was equivalent to the YP187/188AA mutation in mouse Uhrf1. (**b**) Diagram illustrating the *in vitro* reconstituted nucleosome with histone octamers containing with (or without) H3K9me2/3-containing H3. The 200 bp 601 sequence with a biotin at one end was used for *in vitro* nucleosome assembly via salt dialysis. (**c**) Verification of the histone composition of *in vitro* reconstituted nucleosomes with different H3K9 methylation status by Commassie blue staining gel. (**d**) Verification of *in vitro* reconstituted nucleosomes with different H3K9 methylation status by gel mobility shift assay. Asterisk marks free DNA. (**e**) The reconstituted nucleosomes were tested for binding to immobilized GST-UHRF1 by pulldown assay. Note only the nucleosomes with H3K9me2 or H3K9me3 bound to GST-UHRF1 but not the control GST. (**f**) Gel mobility assay examining the nucleosomes derived from *in vitro* assembly with histone octamers containing H3 without N-terminal tail or different levels of H3K9 methylation. (**g**) The nucleosomes assembled in (**f**) were immobilized to streptavidin agarose beads and assayed for binding of GST-UHRF1. The Commassie staining gel at lower panel showed the compositions of core histones in corresponding *in vitro* assembled nucleosomes. (**h**) and (**i**) The pulldown was performed as in (**g**) except the GST-UHRF1 YP191/192AA mutant proteins and GST-UHRF1 (aa 1–407) without SRA domain were used, respectively.

As shown in Fig. 3c, the reconstituted nucleosomes contained the stoichiometric H4, H2A, H2B and His-H3. We also verified the degrees of nucleosome assembly by gel mobility shift assay. As shown in Fig. 3d, majority of DNA was assembled into nucleosomes in each reaction. Two distinct bands were observed for each nucleosome assembly, indicating the presence of two distinct populations of positioned nucleosomes. To test the binding of UHRF1 to these reconstituted nucleosomes, we first

immobilized GST-UHRF1 and control GST to glutathione beads and used them to pulldown reconstituted nucleosomes. We examined the binding of nucleosomes by western blot analysis using an anti-H3 antibody. We found that histone H3 from the nucleosomes reconstituted with H3K9me2 or H3K9me3, but not the nucleosomes with H3K9me0, was retained by GST-UHRF1 but not the control GST proteins (Fig. 3e). To exclude the possibility that the observed binding of nucleosomes with H3K9me2 and H3K9me3 in Fig. 3e was actually due to the binding of free histone octamers and also test the role of H3 N-terminal tail in binding, we reconstituted nucleosomes with DNA containing a biotin moiety at one end (Fig. 3f). We performed reverse pulldown experiments by immobilizing reconstituted nucleosomes to streptavidin beads via biotin moiety at DNA. In this experiment we found GST-UHRF1 bound to the reconstituted nucleosomes with H3K9me2 or H3K9me3, but not the nucleosomes with H3K9me0 or no H3 tail (Fig. 3g). Altogether these experiments demonstrate that UHRF1 binds preferentially to nucleosome containing H3K9me2/3.

To test if the binding of H3K9me2/3-containing nucleosomes by UHRF1 is dependent on its TTD domain, we performed nucleosome pulldown assay for UHRF1 YP191/192AA mutant proteins. Unlike the WT UHRF1, this mutant exhibited no preferentially binding of the H3K9me2/3-containing nucleosomes in multiple experiments (Fig. 3h). Furthermore, we tested the nucleosome binding ability of a GST-UHRF1 (aa 1–470) fusion protein with the deletion of SRA domain. The results in Fig. 3i showed this protein exhibited a preferential binding of nucleosomes with H3K9me2/3. Altogether we conclude that UHRF1 can preferentially bind nucleosomes with H3K9me2/3 in a TTD domain-dependent and SRA domain-independent manner.

**Nucleosome positioning affects UHRF1 binding to hemi-mCpGs.** Although UHRF1 binds hemi-methylated DNA *in vitro*[24,35], how nucleosome structure affects the binding of UHRF1 to hemi-methylated DNA has not been investigated. As the effect of nucleosome on binding of transcription factor is likely influenced by nucleosome positioning[45,46], we wished to test the binding of UHRF1 to nucleosomes with hemi-methylated CpG sites located either in the linker or within core nucleosome. To generate *in vitro* reconstituted nucleosome with defined positioning, we made use of the well characterized 146 bp '601' nucleosome positioning sequence[42]. We therefore assembled nucleosomes with the 200 bp fragment containing no hemi-methylated CpG sites, four hemi-methylated CpG sites outside of or within the edge of 601 sequence as illustrated in Fig. 4a. To place the hemi-methylated CpG sites close to the dyad of nucleosome, we also assembled nucleosomes with a 163 bp fragment with the 146 bp 601 sequence in the middle (Nuc-5meC4 and Nuc-5meC5 in Fig. 4a). The successful assembly of these nucleosomes was confirmed by gel mobility shift assay (Fig. 4b). It is noteworthy that while two distinct nucleosome populations were observed for nucleosome assembly reactions with the longer DNA sequence, only one population of nucleosomes was observed for the short 163 bp DNA fragment. These results indicated that nucleosomes assembled with the longer DNA had two differently positioned nucleosome populations, whereas the nucleosomes assembled with the short DNA had only one uniformly positioned nucleosome. To test how nucleosome positioning affects UHRF1 binding of hemi-methylated DNA, we first compared the binding of nucleosomes assembled with longer DNA with immobilized GST-UHRF1. As shown in Fig. 4c, we observed the binding of nucleosomes with hemi-methylated DNA but not the control nucleosome without hemi-methylated DNA. This binding was UHRF1-dependent, as no binding was observed for control GST

(Fig. 4c). Interestingly, reduced binding was observed when four hemi-methylated CpG sites were placed at the edge of core nucleosome (compare Nuc-5meC3 with Nuc-5meC1 and Nuc-5meC2), suggesting that the binding of UHRF1 was suppressed when hemi-methylated CpGs were embedded within nucleosome. To substantiate this observation, we performed reverse pulldown assay by immobilizing assembled nucleosomes to streptavidin agarose beads. As shown in Fig. 4d, we found that GST-UHRF1 bound less efficiently to Nuc-5meC3 than Nuc-5meC1 and Nuc-5meC2, thus supporting the idea that nucleosome structure inhibits the binding of UHRF1 to hemi-methylated DNA.

A caveat for above nucleosome pulldowns was the presence of two distinct nucleosome populations, possibly as a result of different nucleosome positioning. We thus tested the binding of UHRF1 to nucleosomes reconstituted with the shorter 163 bP DNA fragment (Fig. 4a), which exhibited only one population of positioned nucleosomes (Fig. 4b). To better evaluate the effect of nucleosome positioning on binding of UHRF1, two hemi-methylated CpG sites were placed either close to the dyad (Nuc-5meC4) or the edge (Nuc-5meC5) of nucleosome. When these nucleosomes were subjected to GST pulldown experiments together with Nuc-5meC1 and Nuc-5meC3, we found that, similar to Nuc-5meC3, Nuc-5meC5 exhibited a 2–3-fold reduced binding in comparison with Nuc-5meC1. However, Nuc-5meC4 exhibited a further reduced binding (approximately threefold) in comparison with Nuc-5meC3 and Nuc-5meC5 (Fig. 4e). Again the observed binding of nucleosomes was UHRF1-dependent, as no binding was observed for control GST (Fig. 4e). These results indicate that the hemi-methylated CpG sites at the dyad of nucleosomes were less accessible or more severely inhibited for binding of UHRF1 than the sites at the edge of nucleosomes.

**UHRF1 binds nucleosome with hemi-mCpGs better than with H3K9me3.** We next wished to compare how UHRF1 binds nucleosomes with either H3K9me2/3 or hemi-methylated DNA or both. For this purpose, we first assembled nucleosomes with H3K9me0 and H3K9me3 using the 200 bp sequence with or without hemi-methylated CpG sites outside the 601 nucleosome positioning sequence, thus mimicking the presence of hemi-methylated CpGs in linker region of nucleosome (Fig. 5a). These nucleosomes were immobilized to streptavidin agarose beads and used for pulldown of GST-UHRF1. As shown in Fig. 5b, GST-UHRF1 bound much more efficiently to the nucleosomes with hemi-methylated CpGs than that with H3K9me3 (Fig. 5b, compare lane 5 with lane 4). Furthermore, GST-UHRF1 bound better to the nucleosomes with both hemi-methylated DNA and H3K9me3 (Fig. 5b, compare lane 6 with lanes 5 and 4), suggesting these two modifications act cooperatively to recruit UHRF1. The observed differences in binding of GST-UHRF1 were unlikely due to uneven immobilization of nucleosomes, as Commassie staining revealed the presence of same amounts of core histones in the bead fractions (Fig. 5b, lower panel). We further compared the binding of UHRF1 with *in vitro* reconstituted nucleosomes with H3K9me3 to that with hemi-methylated CpG sites at the edge of nucleosomes (Fig. 5c). As shown in Fig. 5d, again UHRF1 bound more avidly to the nucleosomes with hemi-methylated CpGs at the edge of nucleosomes than that with H3K9me3. These results provide first evidence that although UHRF1 can bind nucleosomes with either hemi-methylated DNA or H3K9me3, it binds with higher affinity to nucleosomes with hemi-methylated DNA, thus providing a molecular explanation that UHRF1-mediated DNA maintenance methylation is largely independent of its H3K9me2/3-binding activity.

Finally, we wished to determine if the presence of H3K9me3 could rescue to what extent the binding of UHRF1 to hemi-methylated CpGs at the dyad of nucleosome. To this end, we

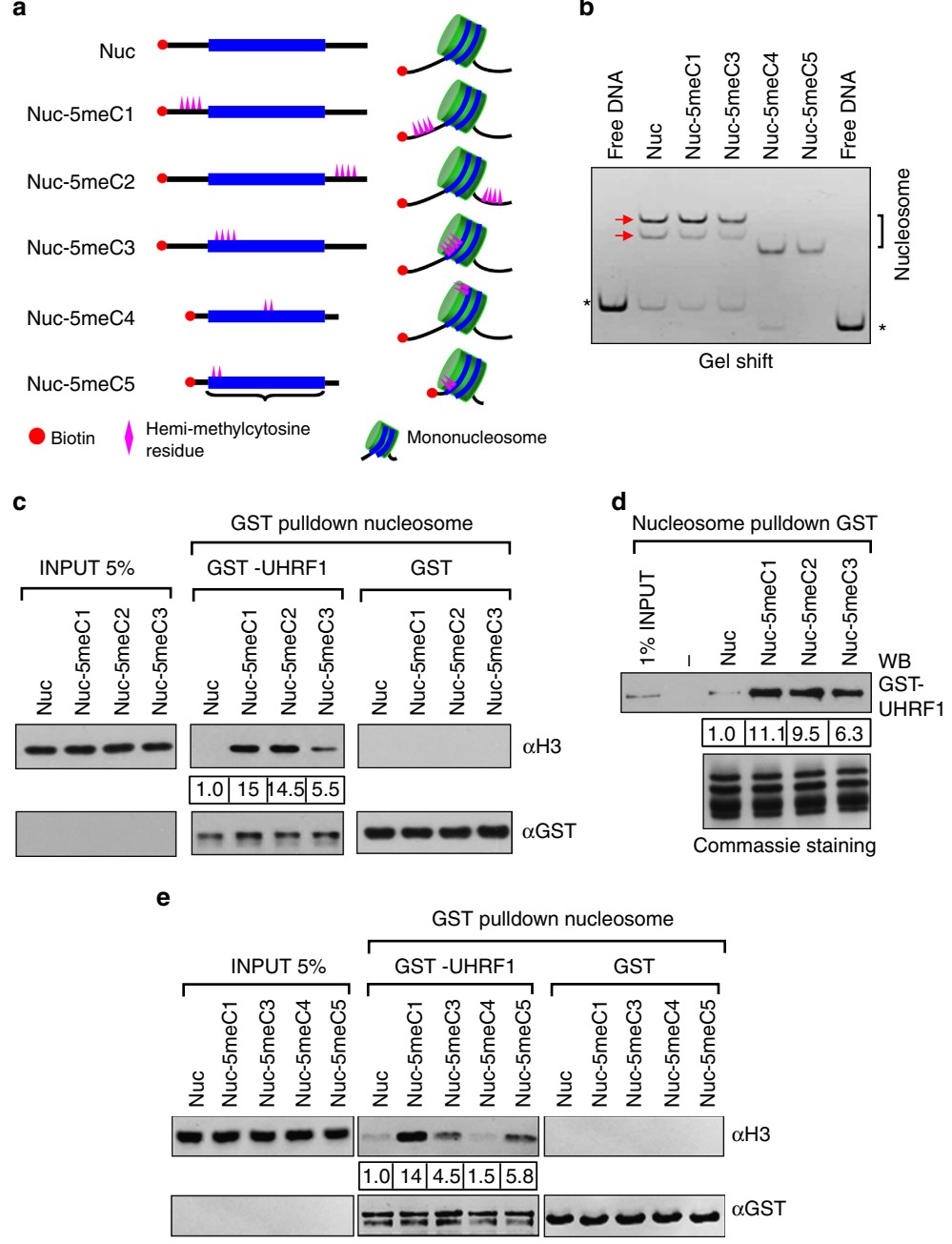

**Figure 4 | UHRF1 binds nucleosomes with hemi-methylated DNA and the binding is influenced by the nucleosome positioning of hemi-methylated CpGs.** (**a**) Diagram illustrating six different types of *in vitro* reconstituted nucleosomes with hemi-methylated CpG at distinct positions. The DNA fragments for Nuc, Nuc-5meC1, Nuc-5meC2 and Nuc-5meC3 were 200 bp with the 146 bp 601 sequence at the middle, whereas the DNA fragments for Nuc-5meC4 and Nuc-5meC5 were 163 bp. The relative positions and number of hemi-methylated CpGs were shown as.. (**b**) Gel mobility shift assay verified the *in vitro* reconstituted nucleosomes. Asterisk marks free DNA. (**c**) Immobilized GST-UHRF1 was used to pulldown *in vitro* reconstituted nucleosomes with hemi-methylated CpGs outside or within the edge of nucleosomes. (**d**) *In vitro* reconstituted nucleosomes as in (**c**) were immobilized to streptavidin agarose beads and used to pulldown GST-UHRF1 proteins. (**e**) Immobilized GST-UHRF1 was used to pulldown *in vitro* reconstituted nucleosomes with hemi-methylated CpGs outside, within the edge or near the dyad of nucleosomes.

assembled the short form of DNA with hemi-methylated CpGs at the edge (5meC5) or dyad (5meC4) into nucleosomes (Fig. 5e) and carried out GST-UHRF1 pulldown assay (Fig. 5f). While nucleosome structure significantly inhibited the binding of UHRF1 (compare H3K9Me0-5meC5 with H3K9Me0-5meC4), the presence of H3K9me3 restored the binding of UHRF1 by ~50%. Thus, the presence of H3K9me3 may act to recruit UHRF1 when hemi-methylated CpGs were blocked by positioned nucleosomes.

## Discussion

In this study we have generated a KI mouse model which specifically inactivates the H3K9me2/3-binding activity of Uhrf1. This KI mouse model has allowed us to determine the precise role of Uhrf1's H3K9me2/3-binding activity in DNA maintenance methylation in mouse. Our data clearly demonstrate that the H3K9me2/3-binding activity of Uhrf1 promotes but is not essential for bulky DNA methylation in mouse. Thus, unlike *Neurospora* and *Arabidopsis* in which part of our entire DNA

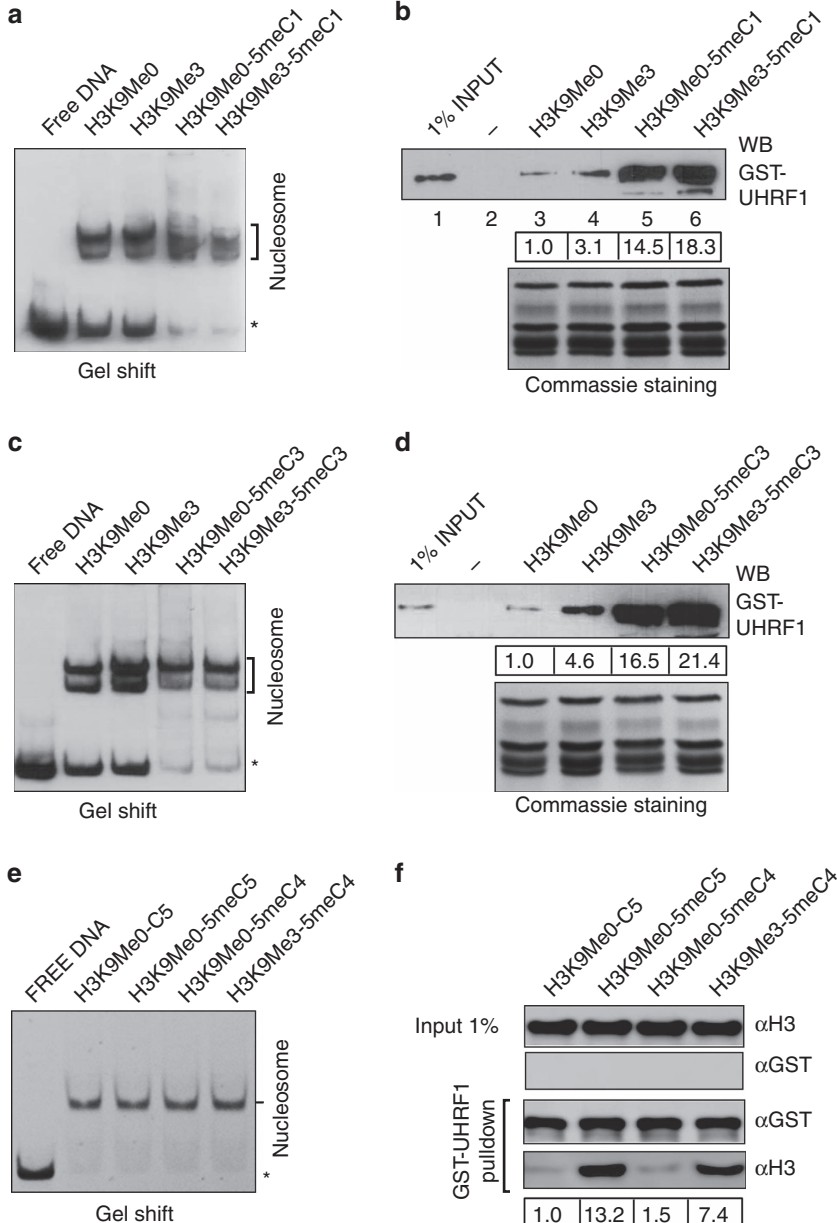

**Figure 5 | H3K9me3 and hemi-methylated CpGs cooperatively enhance the binding of UHRF1 to nucleosomes.** (**a**) Gel mobility shift assay showing the nucleosomes reconstituted with histone octamers containing with or without H3K9me3 and 200 bp DNA fragment with or without hemi-methylated CpGs outside of the nucleosome. (**b**) *In vitro* reconstituted nucleosomes as in (**a**) were immobilized to streptavidin agarose beads and used to pulldown GST-UHRF1 proteins. (**c**) Gel mobility shift assay showing the nucleosomes reconstituted with histone octamers containing with or without H3K9me3 and 200 bp DNA fragment with or without hemi-methylated CpGs near the edge of nucleosome. (**d**) *In vitro* reconstituted nucleosomes as in (**c**) were immobilized to streptavidin agarose beads and used to pulldown GST-UHRF1 proteins. (**e**) Gel mobility shift assay showing the nucleosomes reconstituted with 163 bp DNA fragments with hemi-methylated CpGs at the edge or near the dyad of nucleosomes and histone octamers with or without H3K9me3. (**f**) GST-UHRF1 was immobilized and used for pulldown of *In vitro* reconstituted nucleosomes as in (**e**). Note that the binding of UHRF1 to the hemi-methylated CpGs near the dyad of nucleosome was substantially inhibited in comparison with the sites near the edge of nucleosome and that presence of H3K9me3 in nucleosome partially restored the binding of UHRF1.

methylation pathways are dependent on H3K9 methylation[1,10,47,48], DNA maintenance methylation in mammalian cells by UHRF1-DNMT1 axis, which is responsible for majority of DNA methylation in mammalian cells is, to a large extent, independent on H3K9 methylation.

As two major mechanisms for epigenetic regulation, DNA methylation and histone modifications are expected to crosstalk with each other. Although accumulating evidence supports crosstalk between DNA methylation and histone modifications

in *Neurospora*, plant and mammals, the underlying mechanisms actually differ considerably. In *Neurospora,* DNA methylation is strictly dependent on H3K9 methylation because the DNA methyltransferase DIM-2 is recruited to chromatin by its interaction with a HP1-like protein, which binds chromatin via binding H3K9me2 generated by H3K9 methyltransferase DIM-5 (refs 10–12). In *Arabidopsis,* two of the three DNA methylation pathways, namely the CMT2–CMT3 pathway and RNA-directed DNA methylation pathway are dependent

on H3K9 methylation[9]. The DNA methyltransferase CMT2 preferentially methylates unmethylated CHH and CHG, whereas CMT3 prefers hemimethylated CHG sites. Both CMT2 and CMT3 are recruited to chromatin by binding H3K9me2 through a pair of BAH and chromodomain embedded within themselves[15]. Loss of KRYPTONITE (KYP, also known as SUVH4), a H3K9 mono- and di-methyltransferase leads to reduction in both H3K9me2 and CHH/CHG methylation levels[13]. Interestingly, CHG methylation also promotes H3K9 methylation by KYP, because KYP contains an SRA domain that binds DNA with methylated CHG sites[49]. Thus a positive-feedback loop exists between KYP and CMT3 that efficiently maintains H3K9 methylation and CHG methylation in heterochromatic regions. However, it is noteworthy that CG methylation in *Arabidopsis* is catalysed by MET1, a functional equivalent of mammalian DNMT1. Like maintenance methylation in mammals, CG methylation by MET1 is dependent on the mammalian UHRF1 homologues VIM proteins[50]. Interestingly, current evidence indicates that unlike the CMT2–CMT3 pathway and RNA-directed DNA methylation pathway, CG methylation by MET1 in *Arabidopsis* is actually independent of H3K9 methylation[9]. This is likely due to the absence of a TTD domain in the VIM proteins. Thus, H3K9 methylation-dependent DNA methylation in both *Neurospora* and *Arabidopsis* are mediated through either methylated H3K9-binding domains embedded in DNA methyltransferases or accessory proteins such as HP1 whose association with chromatin is dependent on H3K9 methylation.

Our study on how UHRF1 binds to *in vitro* reconstituted nucleosomes provides mechanistic explanation on why the H3K3me2/3 binding activity of UHRF1 is not essential for bulky DNA maintenance methylation in mammalian cells. Although UHRF1 binds nucleosomes with both H3K9me3 and DNA methylation cooperatively (Fig. 5), UHRF1 can bind nucleosomes with either H3K9me2/3 or DNA methylation (Figs 3 and 4). Furthermore, in our *in vitro* assay UHRF1 appears to bind with a higher affinity to nucleosomes with DNA methylation than that with H3K9 methylation (Fig. 5), although the exact binding affinity for the H3K9me3- or hemi-CpG- containing nucleosomes remain to be determined. Our *in vitro* data are consistent with previous observations that the UHRF1 mutants with impaired H3K9me2/3-binding activity or hemi-methylated DNA maintain a variable degree of heterochromatin association[35,51,52]. Thus, unlike the DNA methylation accessory protein HP1 in *Neurospora* and DNA methyltransferases CMT2/CMT3 in *Arabidopsis*, which are recruited to chromatin via binding of methylated H3K9, our study provides evidence that mammalian UHRF1 is most likely recruited to replication forks primarily via binding of hemi-methylated DNA.

Our study also reveals, for the first time, that the binding of UHRF1 to hemi-methylated CpG sites is influenced by nucleosome positioning. UHRF1 binds well to hemi-methylated CpG sites within linker regions and its binding is suppressed more severely when the CpG sites were located at the dyad of a nucleosome than at the edge. To overcome this inhibitory effect of nucleosome, DNA maintenance methylation may either occur before the assembly of nucleosomes during DNA replication or make use of chromatin remodelling activity[53–55]. In this regard, a chromatin remodeler LSH has been shown to be required for efficient DNA methylation in mammals[54]. Thus, future work will investigate if LSH promotes the targeting of UHRF1 to DNA replication forks in S phase of cell cycle.

Our genome scale analysis of DNA methylation by RRBS failed to identity genomic regions whose DNA methylation is strictly dependent on the H3K9me2/3-binding activity of UHRF1. Instead, we found that majority of the DNA methylation sites are affected, with an average reduction of DNA methylation

∼10%. These results suggest that the binding of H3K9me2/3 plays a broad role in maintenance methylation and is not restricted to particular chromatin domains with enriched H3K9me2/3. Alternatively, majority of the regions with CpG methylation in the genome may also be enriched of H3K9me2/3. This view is supported by previous genome wide analysis of histone H3K9 methylation and DNA methylation[56]. Nevertheless, our data argue against the conclusion that UHRF1 association with H3K9 methylation is essential for DNA maintenance methylation. We thus favour a working model in which H3K9 methylation promotes DNA maintenance methylation in mammalian cells via its ability to enhance the recruitment of UHRF1 through the TTD-PHD domains in UHRF1. However, due to the presence of SRA domain and its higher affinity for binding hemi-methylated DNA, UHRF1 appears to be recruited to replication forks primarily through binding of hemi-methylated DNA. This model is also consistent with a recent study showing that the binding of hemi-methylated DNA by SRA opens a closed conformation of UHRF1 to facilitate its binding of H3K9me3 (ref. 57). We believe the combinatorial H3K9me2/3 and hemi-methylated DNA binding activities of UHRF1 have the advantage to allow on one hand the crosstalk between H3K9 and DNA methylation and the other hand more efficient and accurate inheritance of DNA methylation, as the recognition of hemi-methylated CpG sites is likely more efficient and accurate for DNA maintenance methylation than reading of H3K9 methylation. Thus, although DNA methylation and H3K9 methylation are highly conserved epigenetic modifications, the mechanisms for crosstalk between them are divergent from *Neurospora* to *Arabidopasis* and to mammalian cells. Taken together, our study has not only elucidated the precise role of UHRF1's H3K9me2/3-binding activity in DNA methylation but also has provided novel insight into the mechanism of DNA maintenance methylation in mammalian cells.

## Methods

**Generation of the Uhrf1 YP187/188AA KI mice.** To generate Uhrf1 YP187/188AA KI mice, a recombineering-based method was used for targeting vector construction as described previously[58]. In brief, the genomic DNA corresponding to the exon 3–exon 11 of Uhrf1 was first cloned into the retrieval vector pL253 by gap repair mechanism. Next, the Neo gene flanked by FRT sites was inserted into the intron between exon 3 and exon 4. The YP187/188AA mutations were then introduced into the resulting targeting vector by site-directed mutagenesis. The targeting vector was introduced into 129/SV-derived ES cells (R1) through electroporation and the clones with homologous recombination were double selected by using G418 and ganciclovir. About two hundred clones were picked and screened for correct homologous recombination by PCR using the specific primers which recognized Neo sequences. (Forward: 5′-tctgtgggtactgatagtgctcg-3′, Reverse: 5′-tatcgccttcttgacgagttc-3′). The positive ES clones were further verified by DNA sequencing. Then, the heterozygous ES cells were injected into the blastocysts of C56BL/6 mice to produce chimeric mice. The chimeric mice were cross-bred with C57BL/6 mice to generate F1 mice and the heterozygous KI mice were screened by PCR using the primers described above.

All mice were maintained at the Laboratory Animal Center of East China Normal University. All procedures followed guidelines consistent with those developed by the National Institute of Health and the Institutional Animal Care and Use Committee of the East China Normal University. Uhrf1 WT and mutant mice used for experiments have been purified to C57BL/C genetic background.

**HPLC analysis of global DNA methylation.** The measurement of global DNA methylation by HPLC was performed essentially as described[35]. In brief, genomic DNA was prepared from MEFs or various mouse tissues. The RNA-free genomic DNA samples were treated with nuclease P1 (Wako) and cloned alkaline phosphatase (CIAP, NEB) before they were subjected to HPLC (Agilent 2000 Series, Agilent Eclipase XDB-C18) analysis.

**RRBS analysis.** Genomic DNA was prepared from the liver tissues of a 3-month-old WT and KI male mice. To minimize the potential individual differences, genomic DNAs from three mice were pooled together for RRBS analysis. The RRBS analysis was performed as described[59].

**Plasmids and antibodies.** The plasmids for pGEX4T-1-UHRF1 (aa 95–610) and pGEX4T-1-UHRF1 (aa 1–407) was as described[52]. The pGEX4T-1-UHRF1 (aa 95–610)-YP191/192AA was derived from pGEX4T-1-UHRF1 (aa 95–610) by site-directed mutagenesis. Antibodies used in this study were as follows: anti-UHRF1 as described[52], anti-H3 (Abcam, #ab7766), anti-AcH3 (Abcam, #ab47915), anti-H3K27me3 (CST, #9733S), anti-H3K9me3 (Abclonal, #A2360), anti-GST (HuaAn, #EM80701) and anti-GAPDH (AbMart, #M20006).

**Peptide pulldown assay with biotinylated H3 peptides.** H3 tail peptide pulldown assay was performed essentially as described[52].

**Preparation of recombinant UHRF1 proteins, core histones and H3 with H3K9me2 or H3K9me3.** For purification of GST fusion of UHRF1 proteins, GST-tagged UHRF1 (aa 95–610) or other mutants was overexpressed in *Escherichia coli* BL21 CodonPlus (DE3) RIL cells. The expression of proteins was induced with 0.1 mM β-D-1-thiogalactopyranoside (IPTG) at 37 °C for 3 h. The subsequent purification of GST-fusion proteins was essentially as described[52]. The expression and purification of core histones were essentially as described[43]. Preparation of histone H3 with H3K9me2 and H3K9me3 was essentially as described[44].

**Octamer reconstitution and mononucleosome assembly.** For octamer reconstitution, equimolar amounts of individual histones in unfolding buffer (7 M Guanidine HCl, 20 mM Tris–HCl, pH 7.5, 10 mM β-ME) were dialysed into refolding buffer (2 M NaCl, 10 mM Tris–HCl, pH 7.5, 1 mM EDTA, 5 mM β-ME), and purified with Superdex200 column (24 ml) essentially as described[43]. Mononucleosome was assembled using salt-dialysis as described[43].

The non-methylated DNA and the hemi-methylated DNA used for the assembly of mononucleosomes included the 601 positioning sequence were prepared by PCR using biotinylated primers. Primers used in PCR reactions were as follows: 601-M0-F (5′-BIOTIN-TACCGAACGTTCGAACCATGA TGCCGGAT-3′) and 601-M0-R (5′-TACGCGAATTCCAAGCGACACCGG CACT-3′) for Nuc (non-methylated DNA), 601-M1-F (5′-BIOTIN-TACmCp GAAmCpGTTmCpGAACCATGATGCmCpGGAT) and 601-M0-R for Nuc-5meC1 (hemi-methylated DNA), 601-M0-F and 601-M2-R (5′-TAmCpGm CpGAATTCCAAGmCpGACACmCpGGCACT-3′) for Nuc-5meC2 (hemi-methy-lated DNA), 601-M3-F (5′-BIOTIN-TACCGAACGTTCGAACCATGATGCC GGATCCC.

CTGGAGAATCCmCpGGTGCmC pGAGGCmCpGCT CAATTGGTmCpG) and 601-M0-R for Nuc-5meC3 (hemi-methylated DNA), 601-M4-F (5′-BIOTIN-ATCCCCTGGAGAA TCCCGGTGCCGAGGCCGCTCAATTGGTCGTAGACA GCTCT AGCACmCpGCTTAAAmCpG-3′) and 601-M4-R (5′-CCGGCACTGG AACAGGATGTATAT-3′) for Nuc-5meC4(hemi-methylated DNA), 601-M5-F (5′-BIOTIN-ATCCCCTGGAGAATCCmCpGGTGCmCpGG-3′) and 601-M4-R for Nuc-5meC5. The length of DNA fragment was 200 bp for Nuc, Nuc-5meC1, Nuc-5meC2 and Nuc-5meC3 and 163 bp for Nuc-5meC4 and Nuc-5meC5.

**Mononucleosome pulldown and GST pulldown assays.** For pulldown of UHRF1 proteins using immobilized nucleosomes, the mononucleosomes were mixed with streptavidin agarose beads in the binding buffer (20 mM Tris–HCl, pH 7.6, 10% glycerine, 100 mM KCl, 10 mM β-mercaptoethanol, 1 mM dithiothreitol (DTT), 0.5 mM phenylmethylsulphonyl fluoride (PMSF), 50 ng μl$^{-1}$ bovine serum albumin) for 2 h at 4 °C. Mononucleosomes-bound streptavidin agarose beads were washed twice with the binding buffer and then were incubated with recombinant WT or mutant GST-UHRF1 in the binding buffer for 4 h at 4 °C. After incubation, the bound proteins were washed five times with the washing buffer (20 mM Tris–HCl, pH 7.6, 10% glycerine, 100 mM KCl, 10 mM β-mercaptoethanol, 1 mM DTT, 0.5 mM PMSF) and were analysed by Western blotting. The pulldown with GST-UHRF1 proteins were essentially as above except the GST-UHRF1 proteins were immobilized to glutathione agarose beads. The quantification of relative binding was determined by using Image J.

**Statistics.** Statistical significance of global DNA methylation was assessed by the paired Student's tests, mean ± s.e.m. ($n = 3$). The genotype and gender distribution of filial generation mice was appraised by $\chi^2$ test.

**Data availability.** All relevant data are available from the authors on request.

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

## Acknowledgements

We thank members of Wong's laboratory for valuable discussion. This study is supported by the grants to J.W. from the National Science and Technology Major Project 'Key New Drug Creation and Manufacturing Program' of China (2014ZX09507002-002), the Ministry of Science and Technology of China (2015CB910402), the National Natural Science Foundation of China (91419303) and The Science and Technology Commission of Shanghai Municipality (14XD1401700, 11DZ2260300); to G.L. from the National Natural Science Foundation of China (91219202 and 31525013), and the Ministry of Science and Technology of China (2015CB856200).

## Author contributions

G.L. and J.W. conceived and designed the experiments. Q.Z. and W.L. performed nucleosome reconstitution and binding assays. Z.J., C.R. and S.G. made and characterized the knockin mice model. D.X. and W.W. performed HPLC analysis of DNA methylation. Z.H., W.Q., C.H., H.J. and Y.W. performed RRBS analysis of DNA methylation. L.J. contributed to the design and execution of this study. J.W. wrote the manuscript. All authors reviewed and approved the manuscript.

## Additional information

**Competing financial interests:** The authors declare no competing financial interests.

