## [Peer Review File · Nature Communications]

Reviewers' comments:

Reviewer #1 (Remarks to the Author):

The work from Zhao et al. shows that the DNA methylation maintenance complex (UHRF1 and DNMT1) is capable of binding H3K9 methylated nucleosomes but that this mark is not strictly essential for its activity. They performed in vivo and in vitro experiments showing that hemimethylated DNA is a stronger recruiting signal than H3K9me for UHRF1. Moreover the position of the hemimethylated sites within the nucleosome influence the binding of UHRF1 in vitro.

minor comments:

1. In the paragraph describing the RRBS sequencing data, the authors should mention the number of uniquely mapped reads and the number of CpG sites covered by at least 5 reads. Since the unmapped reads and the cytosines with less than 5 reads have not been used in the subsequent analysis, they should not be mentioned in the main text. However table 2A should report all the information.
2. In the main text, the authors say "we identified in the KI mutant 20308 hypomethylated regions", while they actually identified hypomethylated individual sites/CpGs. Please correct.
3. The legend of Figure 2 is not correct. The legend for the real Figure 2C is absent and the subsequent legends are not referred to in the correct figure.

Additional experiments:

I would add an additional experiment similar to the one described in Figure 5, but using the Nuc-5meC4 sequence. The question is: in the absence of an accessible hemimethylated cytosine (close to the dyad), can the presence of H3K9me3 rescue the binding levels of UHRF1-Nuc5meC1/2/3/5? Is this binding stronger than unmethylated DNA wrapped around a H3K9me0 nucleosome?

Reviewer #2 (Remarks to the Author):

The multi-domain protein Uhrf1 is an accessory factor of Dnmt1 and is essential for maintaining DNA methylation. The mechanisms by which Uhrf1 regulates Dnmt1 and DNA methylation have been highly controversial. Specifically, biochemical, cellular and structural studies have implicated almost all the functional domains of Uhrf1 (with the exception of the UBL domain) in its role in DNA methylation. However, the significance of individual functional domains in vivo has not been explored. In this study, Zhao et al. generated knockin (KI) mice carrying two point mutations (Y187A, P188A) in the Tandem Tudor Domain (TTD) that specifically abolish binding to histone H3K9me2/3. The authors showed that the KI mice were normal and fertile, with relatively minor (~10%) decreases in global DNA methylation in various tissues, in contrast to complete deletion of Uhrf1, which results in severe hypomethylation and embryonic lethality. The authors also used in vitro reconstitution experiments to show that Uhrf1 binds nucleosomes with hemi-methylated DNA better than nucleosomes with H3K9me2/3. The results suggest that TTD-mediated H3K9me2/3 binding contributes to, but is not critical for, DNA methylation.

The findings are significant in understanding the mechanisms whereby maintenance DNA methylation is regulated. However, several issues need to be resolved or clarified.

1. The authors showed sequencing data in Fig. 1B and PCR genotyping data in Suppl. Fig. S1. These data do not definitively prove that gene targeting occurred correctly and that there were no random integration events. They need to show Southern blot data (with probes inside and outside of the targeting vector) to rule out those possibilities. They also need to clone the full-length Uhrf1 cDNA

from homozygous KI cells and then sequence it to confirm that all Uhrf1 transcripts have the mutations.

2. According to Fig. 1A, the Neo cassette was not removed in the mutant allele. Even though that did not seem to have a major impact on Uhrf1 protein level (Fig. 1D), minor effect cannot be excluded. That makes one wonder whether the ~10% decrease in DNA methylation was due to the mutations or slight decreases in Uhrf1 protein. I'd suggest that the Neo cassette be removed (by Flpe) either in mice or in cells to clarify the issue.

3. The authors concluded that the mechanisms underlying the crosstalk between histone modifications and DNA methylation in mammalian cells are distinct from in *Neurospora* and plants. That claim is way over-stretched and inaccurate, because regulation of DNA methylation by H3K9 methylation has long been demonstrated in mammalian cells (for example, see Lehnertz et al. *Curr Biol* 2003), although the regulation appears to be more complex than in fungi and plants.

4. Minor issues: TTD was misspelled as TDD in several places. In Fig. 1, the figure panels and the legend do not match. Specifically, Fig. 1C shows the genotypes of pups, which is not included in the figure legend, and the subsequent panels are all off.

Reviewer #3 (Remarks to the Author):

Review of the manuscript: "Dissecting the Precise Role of Histone H3 K9 Methylation in Crosstalk with DNA Maintenance Methylation in Mammalian Cells "

Feb., 2016

Overview:

Zhao and colleagues performed a thorough study of H3K9me3-UHRF1 interaction and the in vivo consequences of its disruption by generating a knock-in mouse with two amino acids substitutions that rendered UHRF1 unable to bind H3K9me3. Authors show that the consequence of such disruption in mice is about 10% decrease in global methylation level, a result contrary to the predominant model in the field. Zhao et al., performed proper biochemical experiments with reconstituted nucleosomes and concluded that UHRF1 binds more efficiently to nucleosomes with hemi-methylated cytosine rather than H3K9me3. The key experiment of this paper (Fig. 5B, D) clearly shows that UHRF1 binding to nucleosomes predominantly depends on 5mC, and is independent of H3K9 methylation. Nucleosome reconstitution experiments also provide important in vitro understanding of the nucleosome structure effect on binding of UHRF1 and recognition of 5mC in the context of nucleosomes.

Minor points:

In the introduction should cite studies about H3K4me0 and DNMT3L (Ooi et al., *Nature* 2007). Many typos are found in the text.

Conclusion

REVIEWERS' COMMENTS:

Reviewer #1 (Remarks to the Author):

I am satisfied with the author's responses to my comments.

Reviewer #2 (Remarks to the Author):

All the issues raised by me have been satisfactorily addressed in the revised manuscript.

Point-to-Point Response to Reviewer's Comments

Reviewer #1 (Remarks to the Author):

First we want to thank this reviewer for the insightful and constructive comments that helped us to improve our manuscript.

Minor comments:

1. In the paragraph describing the RRBS sequencing data, the authors should mention the number of uniquely mapped reads and the number of CpG sites covered by at least 5 reads. Since the unmapped reads and the cytosines with less than 5 reads have not been used in the subsequent analysis, they should not be mentioned in the main text. However table 2A should report all the information.

Answer: We have updated the table in Figure 2A by adding a column "CpG sites w/ >5 reads" as suggested.

2. In the main text, the authors say "we identified in the KI mutant 20308 hypomethylated regions", while they actually identified hypomethylated individual sites/CpGs. Please correct.

Answer: Thanks and we have now changed "regions" to "CpGs" to be precise.

3. The legend of Figure 2 is not correct. The legend for the real Figure 2C is absent and the subsequent legends are not referred to in the correct figure.

Answer: Many thanks for your critical reading. We have now added the missing Figure 2C legend as following: "(C) The distribution of CpGs with different percentage of methylation levels in WT and Uhrf1 KI. 2654195 CpGs covered by both samples were analyzed here. X axis is the percentage of DNA methylation, Y axis is CpG counts in million."

Additional experiments:

I would add an additional experiment similar to the one described in Figure 5, but using the Nuc-5meC4 sequence. The question is: in the absence of an accessible hemimethylated cytosine (close to the dyad), can the presence of H3K9me3 rescue the binding levels of UHRF1-Nuc5meC1/2/3/5? Is this binding stronger than unmethylated DNA wrapped around a H3K9me0 nucleosome?

Answer: We have performed this additional experiment as suggested. The results in new Figure 5 E and F showed that, while the presence of H3K9me0 nucleosome strongly inhibited the binding of UHRF1 to hemi-methylated CpGs close to the dyad (H3K9Me0-5meC4), the presence of H3K9me3 nucleosome partially rescued the binding and this binding is stronger than binding of H3K9me0 nucleosome without DNA methylation.

Reviewer #2 (Remarks to the Author):

We want to thank this reviewer for the positive comments such as "The findings are significant in understanding the mechanisms whereby maintenance DNA methylation is regulated". We also want to thank this reviewer for the insightful comments that helped us to improve our manuscript.

1. The authors showed sequencing data in Fig. 1B and PCR genotyping data in Suppl. Fig. S1.

These data do not definitively prove that gene targeting occurred correctly and that there were no random integration events. They need to show Southern blot data (with probes inside and outside of the targeting vector) to rule out those possibilities. They also need to clone the full-length Uhrf1 cDNA from homozygous KI cells and then sequence it to confirm that all Uhrf1 transcripts have the mutations.

Answer: We agree with the reviewer that the sequencing data in Fig. 1B and Fig. S1 do not definitively prove that gene targeting occurred correctly and that there were no random integration events in the knockin mice. We agree a genomic Southern blot data would prove if there is any random integration event. Due to our lack of facility for radioactive materials, we addressed this issue by PCR detection of genomic DNA using several pairs of primers specific for targeting vector. As shown in Supplementary Figure. S1 for reviewer only, none of the three pairs (#2-4) of targeting vector-specific primers gave rise to any detectable PCR product, whereas they generated strong products with expected sizes from the control targeting vector. As a control, the primer pair #1 used for genotyping generated the expected 607bp PCR product from the KI but not WT mice, further demonstrating the correct homologous recombination event in the KI mice. Together these data demonstrate there is lack of targeting vector sequence in the KI mice, thus excluding the possibility for the presence of random integrated targeting vector.

Also as suggested by the reviewer, we have cloned and sequenced the entire Uhrf1 cDNA from the KI mice and the result in Supplementary Figure. S2 for reviewer only indicates there is no additional mutation except the expected TATCCA to GCTGCA mutations introduced by knock-in.

2. According to Fig. 1A, the Neo cassette was not removed in the mutant allele. Even though that did not seem to have a major impact on Uhrf1 protein level (Fig. 1D), minor effect cannot be excluded. That makes one wonder whether the ~10% decrease in DNA methylation was due to the mutations or slight decreases in Uhrf1 protein. I'd suggest that the Neo cassette be removed (by Flpe) either in mice or in cells to clarify the issue.

Answer: We believe the ~10% decrease in DNA methylation was due to the mutations but not potential difference in Uhrf1 protein for the following reason. In addition to the knock-in mouse model, we have also generated a knockout mouse model for Uhrf1. While the homozygous KO mice are embryonic lethality as reported previously, the heterozygous mice are viable and show no gross phenotype. Furthermore, we found that the levels of DNA methylation in livers for heterozygous mice are in no difference from the wild-type mice, despite of a small reduction of Uhrf1 proteins in the livers of heterozygous mice (Xiaoya Duan and Jiemin Wong, unpublished data). Since the amount of Uhrf1 proteins are sufficient for DNA maintenance methylation even in the heterozygous KO mice, the observed reduction of DNA methylation in KI mice is extremely unlikely due to the potential minor difference (if any) in the levels of Uhrf1 proteins in KI mice.

3. The authors concluded that the mechanisms underlying the crosstalk between histone modifications and DNA methylation in mammalian cells are distinct from in *Neurospora* and plants. That claim is way over-stretched and inaccurate, because regulation of DNA methylation by H3K9 methylation has long been demonstrated in mammalian cells (for example, see Lehnertz et al. *Curr Biol* 2003), although the regulation appears to be more complex than in fungi and plants.

Answer: Although some literatures including the one by Lehnertz et al. have shown regulation of

DNA methylation by H3K9 methylation, the focus of our study is whether DNA methylation or certain pathway(s) of DNA methylation in mammals is dependent on H3K9 methylation, as in the cases of Neurospora and plants. We have now cited the work by Lehnertz in the introduction. Although it is not clear to what extent DNA methylation by Dnmt3b is regulated by H3K9 methylation, we think there is no conflict between his and our study.

4. Minor issues: TTD was misspelled as TDD in several places. In Fig. 1, the figure panels and the legend do not match. Specifically, Fig. 1C shows the genotypes of pups, which is not included in the figure legend, and the subsequent panels are all off.

Answer: Many thanks and we have corrected the misspelling and errors in Fig. 1 legend.

Reviewer #3

We want to thank this reviewer for no major criticism.

Minor points:

In the introduction should cite studies about H3K4me0 and DNMT3L (Ooi et al., Nature 2007).

Many typos are found in the text.

Answer: We thank this reviewer for reminding us to cite the important work by Ooi et al, which is now cited. We have also spent significant effort to correct typos in the text.